# TriGuardFL: Triple-Step Byzantine-Robust Federated Learning against Model Poisoning Attacks

## Abstract

Federated learning's (FL) distributed architecture is promising, yet it is vulnerable to model poisoning attacks that degrade global model accuracy. Existing defense strategies typically compare the locally updated gradients of clients and exclude or down-weight those exhibiting substantial deviations. However, these strategies may become ineffective when the clients' datasets are heterogeneous. In this paper, we propose `TriGuardFL`, a novel triple-step defense framework that robustly discriminates malicious actors from benign and non-IID clients. First, we employ a cosine-similarity-based filter to identify suspicious clients. Second, a fine-grained secondary evaluation assesses their performance using a small class-stratified dataset. By analyzing class-wise performance differences, it can discern whether a divergent update stems from a malicious attack or data heterogeneity. Finally, a Bayesian reputation model is integrated to manage the uncertainty of detection and enhance the long-term robustness. Extensive case studies on two benchmark datasets and three representative model poisoning attacks demonstrate that `TriGuardFL` outperforms existing methods in mitigating the impact of model poisoning attacks.

The source code is available in the supplementary material to ensure reproducibility and encourage further research.

## 1 Introduction

Federated Learning (FL), a machine learning structure in which clients collaboratively train neural networks to obtain optimal global parameters without sharing their private training samples, has attracted wide attention in recent years (Li et al., 2020; Beltrán et al., 2023; T Dinh et al., 2020; Sun et al., 2023; Guan et al., 2024). However, its distributed nature makes FL inherently vulnerable to model poisoning attacks, where malicious clients attempt to corrupt the global model by submitting deliberately crafted updates (Li et al., 2022; Shejwalkar et al., 2022; Bhagoji et al., 2018). These attacks are broadly classified as untargeted, aiming to degrade overall model accuracy, and targeted, designed to compromise performance on specific tasks, e.g., backdoor attacks (Bagdasaryan et al., 2020; Fang et al., 2020). While both are significant threats, untargeted attacks pose a fundamental challenge to the stability of the collaborative training process and are therefore the focus of this paper.

Byzantine-robust computing has been widely adopted to counter these threats, thanks to their ability to provide strong security guarantees against arbitrary and malicious client behavior without assuming specific attack strategies (Singh et al., 2025; Cao et al., 2021; Yazdinejad et al., 2024). Existing Byzantine-robust defenses against model poisoning are generally categorized into three main types: (i) robust aggregation rules that filter statistical outliers (Blanchard et al., 2017; Yan et al., 2023; Jia et al., 2022; Ma et al., 2022), (ii) server-side validation using a trusted dataset (Cao et al., 2021), and (iii) client-side defense mechanisms (Fang et al., 2024; Sun et al., 2021). These efforts show promise in mitigating Byzantine attacks; however, they continue to face significant challenges in more complex and critical scenarios. For instance, robust aggregation rules are vulnerable to sophisticated attacks that craft malicious parameters to be numerically close to benign ones, thus evading outlier detection (Baruch et al., 2019). Furthermore, distance-based judgments of "closeness" are often distorted by the varying parameter scales across different dimensions of a neural network. More important, these defenses struggle in non-IID settings, where the natural parameter divergence among

benign clients with different data distributions effectively masks malicious updates, rendering the defense unreliable.

We propose `TriGuardFL`, a novel defense architecture designed to address the critical challenge of existing methods in non-IID settings by robustly distinguishing malicious updates from those deviations due to heterogeneous client data. Our approach is built on a key insight: while malicious clients tend to degrade performance uniformly, benign non-IID clients exhibit a distinct statistical signature of high variance in their per-class performance. `TriGuardFL` is a three-stage system designed to detect this signature. The server first identifies potential malicious clients by comparing the parameters of each client with the aggregated one via cosine similarity, and the rest are treated as benign clients in that epoch. Then, considering the training process of non-IID samples with multiple classes of labels on the client side, the server compares the FL model output of potential malicious clients in each class with the aggregated parameter of benign clients. For a target client, if the optimal FL model outputs among different classes of samples based on the objective function are significantly different from the other FL model outputs among different classes of samples and also better than those of potential benign clients in at least one class, the client may be treated as benign ones with non-IID data. Otherwise, the client would be labeled malicious. This process is further enhanced by a Bayesian reputation model to track client behavior over time to improve robustness.

We demonstrate the effectiveness of this approach through extensive experiments on the Fashion-MNIST and CIFAR-10 datasets. The results show that `TriGuardFL` significantly outperforms the existing state-of-the-art Byzantine-robust rules, particularly in challenging non-IID scenarios where they falter. This suggests that explicitly modeling the characteristics of non-IID data, rather than treating all deviations as attacks, is a more robust path forward for safeguarding FL.

## 2 BACKGROUND AND RELATED WORK

### 2.1 FEDERATED LEARNING

Consider an FL framework composed of a central server and a set of $N$ clients. Each client $i \in \{1, \dots, N\}$ has its local dataset $\mathcal{D}_i$. The objective of the Federated Averaging `FedAvg` (Zhong et al., 2021) is defined as follows:

$$\min_{\boldsymbol{w}} F(\boldsymbol{w}) \triangleq \sum_{i \in \mathcal{N}} \frac{|\mathcal{D}_i|}{|\mathcal{D}|} F_i(\boldsymbol{w}_i), \tag{1}$$

where $\boldsymbol{w} \in \mathbb{R}^d$ represents the global model vector, $F(\cdot)$ and $F_i(\cdot)$ stand for the global objective and the local objective of client $i$, respectively, $\mathcal{N}$ denotes the set of all clients. $\mathcal{D} \triangleq \cup_{i=1}^N \mathcal{D}_i$ denotes the union of all clients' datasets, which contains several classes and $\mathcal{C}$ stands for the set of all classes.

For the global and localvobjective functions $F(\cdot)$ and $F_i(\cdot)$, we have the following assumptions.

**Assumption 2.1.** Each local objective function $F_i$ is L-smooth, i.e., $\forall \boldsymbol{w}, \boldsymbol{w}', F_i(\boldsymbol{w}) \leq F_i(\boldsymbol{w}') + (\boldsymbol{w} - \boldsymbol{w}')^T \nabla F_i(\boldsymbol{w}) + \frac{L}{2}||\boldsymbol{w} - \boldsymbol{w}'||_2^2$.

**Assumption 2.2.** Each local objective function $F_i$ is $\mu$-strongly convex, i.e., $\forall \boldsymbol{w}, \boldsymbol{w}', F_i(\boldsymbol{w}) \geq F_i(\boldsymbol{w}') + (\boldsymbol{w} - \boldsymbol{w}')^T \nabla F_i(\boldsymbol{w}) + \frac{\mu}{2}||\boldsymbol{w} - \boldsymbol{w}'||_2^2$.

**Assumption 2.3.** The expected squared norm of the gradients is uniformly bounded, i.e., $\mathbb{E}||\nabla F_i(w_{i,t})||^2 \leq G^2$ for $i = 1, ..., N$ and $t = 1, 2, ..., T$.

The objective function (1) is optimized using iterative client-server communication and training. At each epoch $t$, client $i$ updates its local parameters via gradient descent (GD):

$$\boldsymbol{w}_{i,t}^{k+1} = \boldsymbol{w}_{i,t}^k - \eta_{i,t} \nabla F_i(\boldsymbol{w}_t), \tag{2}$$

where $\eta_{i,t}$ is the learning rate of client $i$ in epoch $t$, and $\nabla F_i(\cdot)$ represents the gradient of the local objective function computed on client $i$'s data. After $K$ rounds of local updates, client $i$ sends its undated model weights $\boldsymbol{w}_{i,t}^K$ to the server.

The server then aggregates the updated model weights by computing a weighted average (`FedAvg`), where each model's contribution is proportional to the size of its client's local dataset. To improve

training efficiency, the server can aggregate models from a randomly selected subset of clients.

$$\boldsymbol{w}_t = \sum_{i \in \mathcal{N}_t} \frac{|\mathcal{D}_i|}{\sum_{i \in \mathcal{N}_t} |\mathcal{D}_i|} \boldsymbol{w}_{i,t}^K, \tag{3}$$

where $\mathcal{N}_t$ denotes the set of selected clients at epoch $t$. A special case is $\mathcal{N}_t = \mathcal{N}$, i.e., all clients involve the model aggregation at epoch $t$. After data aggregation, the server sends the weighted average parameters back to all clients as the initial value of epoch $t + 1$ in (2).

## 2.2 DEGREE OF NON-IID

Let $\mathcal{C}$ stands for the set of all classes. In each dataset $\mathcal{D}_i$, if the sample proportions among different classes are the same and cover all classes in $\mathcal{C}$, then the samples in different datasets are IID; otherwise, $\mathcal{D}_i$ is non-IID. Let $F^*$ and $F_i^*$ denote the optimal global objective and the optimal local objective of the client $i$, respectively. The non-IID datasets impact on the FL model is measured by $\Gamma \triangleq F^* - \sum_{i \in \mathcal{N}} \frac{|\mathcal{D}_i|}{|\mathcal{D}|} F_i^*$. If $\mathcal{D}_i$ is IID for all clients, $\Gamma$ is zero; otherwise, $\Gamma$ is nonzero.

## 2.3 MODEL POISONING ATTACKS

As for attack models, we adopt three state-of-the-art model poisoning attacks: Min-Sum (Shejwalkar & Houmansadr, 2021), Min-Max (Shejwalkar & Houmansadr, 2021), and LIE (Baruch et al., 2019). Following previous studies (Baruch et al., 2019; Fang et al., 2020; Shejwalkar & Houmansadr, 2021; Yan et al., 2023), we consider two cases regarding the prior knowledge of malicious clients: (a) Full: malicious clients know the updated gradients of benign clients; and (b) Partial: malicious clients cannot access the updated gradients of benign clients. For consistency of reading, we use the notation from Shejwalkar & Houmansadr (2021); Baruch et al. (2019).

- **Min-Max** attack constructs a malicious gradient $\nabla^m$ by adding a scaled perturbation vector $\gamma \nabla^p$ to its calculated average gradient $\nabla^b$, i.e., $\nabla^m \triangleq \nabla^b + \gamma \nabla^p$. Subsequently, the scaling factor $\gamma$ is optimized to ensure that the distance between the compromised gradient and the benign average does not exceed the maximum observed distance between benign gradients. This strategy allows the attack to evade detection by methods that mainly compare gradient distances.

- **Min-Sum** generates the malicious gradient $\nabla^m \triangleq \nabla^b + \gamma \nabla^p$, while optimizing $\gamma$ to maximize the sum of the square of distances between the benign and compromised gradients, while not exceeding the sum of the square of distances between any pair of benign gradients.

- **LIE** generates the attack vectors that are near the averaging model parameters, which are easily confused with normal parameters. Specifically, attackers calculate the mean $\mu$ and standard deviation $\sigma$ of benign model parameters according to their prior knowledge and calculate the maximum bias $z^{\max}$ based on the number of benign and malicious clients. The corrupted model parameters are then updated to $\mu - z^{\max} \sigma$ to maximize the poison effect while keeping the attack less noticeable.

## 2.4 BYZANTINE-ROBUST ALGORITHMS

While `FedAvg` and its variants are widely used in FL, they are inherently vulnerable to model poisoning attacks. These attacks can degrade the global model's performance and prevent convergence. To mitigate this threat, a variety of Byzantine-robust aggregation algorithms have been proposed. We outline several representative defense mechanisms below.

- **DeFL** (Yan et al., 2023). The server identifies malicious clients by identifying statistical outliers. It compares each client's update to the others and flags those that deviate significantly. A Bayesian model is then used to assign aggregation weights, effectively discarding the identified malicious updates by setting their weights to zero.

- **FLDetector** (Zhang et al., 2022). This approach relies on historical data to predict the expected model update for each client in a given round. The server measures the consistency between the submitted updates and its predictions. A client is flagged as malicious if its updates consistently deviate from the predicted behavior over multiple epochs.

- **FLTrust** (Cao et al., 2021). Different from the above solutions, for FLTrust, the server owes a clean dataset to compute a trusted "golden" model update in each round. It then calculates the cosine similarity between this trusted update and each client's submitted update. This similarity score serves as the client's aggregation weight, thereby down-weighting clients whose updates diverge from the server's trusted direction.

- **FL-WBC** (Sun et al., 2021). This method analyzes the vulnerability of the parameter using second-order derivative (Hessian) information from the local objective function. Identifies model parameters that are highly sensitive to manipulation and perturbs their updates with Laplace noise to mask the impact of potential poisoning attacks.

- **Multi-Krum** (Blanchard et al., 2017). Multi-Krum scores each client's update based on the sum of its squared Euclidean distances to its $n - f - 2$ nearest neighbors (where $n$ is the total number of clients and $f$ is the maximum number of attackers). Instead of selecting only the single best-scoring update as Krum does, Multi-Krum selects the $m$ updates with the lowest scores and averages them to produce the aggregated model.

While effective under IID settings, current defenses struggle to distinguish between malicious and benign non-IID clients, posing a key challenge for Byzantine-robust FL in non-IID scenarios.

## 3 THE PROPOSED METHOD

`TriGuardFL` proposes method employs a three-step guard mechanism. The first step uses cosine similarity detection to identify clients whose model parameters exhibit low similarity to the global average. The second step performs a statistical significance test to determine whether a client possesses unique data samples. The third step incorporates a robustness enhancement strategy to tolerate potential false negatives. The detailed system design is presented as follows.

### 3.1 COSINE-SIMILARITY-BASED SHORTLIST FOR MALICIOUS CLIENTS

*Cosine similarity* is a tool to measure the difference between two vectors. Here, we adopt cosine similarity to compare the aggregated data with parameters sent from each client, and shortlist potential malicious clients.

At each epoch, the server aggregates the collected data using (3). We refer to this process as the initial aggregation and denote the data initially aggregated at epoch $t$ by $\boldsymbol{w}_t'$, i.e., $\boldsymbol{w}_t' = \sum_{i \in \mathcal{N}_t} \frac{|\mathcal{D}_i|}{\sum_{i \in \mathcal{N}_t} |\mathcal{D}_i|} \boldsymbol{w}_{i,t}^K$. Then, we compute the cosine similarity as follows:

$$\cos \theta_{i,t} = \frac{\boldsymbol{g}_{i,t} \cdot \boldsymbol{g}_t'}{\|\boldsymbol{g}_{i,t}\|_2 \|\boldsymbol{g}_t'\|_2}, \tag{4}$$

where $\cos \theta_{i,t}$ represents the similarity between the two vectors,

$$\boldsymbol{g}_{i,t} \triangleq \boldsymbol{w}_{i,t}^K - \boldsymbol{w}_{t-1}/\eta_{i,t}, \ \boldsymbol{g}_t' \triangleq \boldsymbol{w}_t' - \boldsymbol{w}_{t-1}/\eta_{i,t}, \tag{5}$$

. If $\cos \theta_{i,t}$ is below the threshold, the client $i$ may be malicious. The server computes $\boldsymbol{w}_t'$ and $\cos \theta_{i,t}$ for each client using the method mentioned above. As a result, the malicious clients shortlisted are those whose $\cos \theta_{i,t}$ values fall below the threshold. Here, we set a threshold $\delta_1$. For each client $i$, if $\cos \theta_{i,t} < \delta_1$, it is considered a potential malicious client and should be further examined. The value of $\delta_1$ represents the strictness of detection, the larger value means more clients may be shortlisted as potential malicious ones. Let $\mathcal{M}_t'$ and $\mathcal{B}_t'$ denote the sets of shortlisted and nonshortlisted clients, respectively.

### 3.2 SIGNIFICANCE-TEST-BASED MALICIOUS CLIENTS DETECTION

In the above subsection, the server has shortlisted potential malicious clients. However, benign clients whose cosine similarity falls below the threshold may also be included. This is particularly relevant in non-IID scenarios, where a client may hold one or more classes of training data not held by other clients. As a result, its updated parameters may differ significantly from those of others, leading to being classified as a potential malicious client. Therefore, distinguishing benign clients from the shortlist is critical to reducing false alarms.

To address this problem, we propose an algorithm based on objective function output value comparison. First, we make the following assumption about the server:

**Assumption 3.1.** The server has access to a clean few-shot clean dataset $\mathcal{D}'$ that contains all the classes present in the training data of clients.

Note that in Assumption 3.1, the clean dataset is a few shots in size, which complies with the principle of FL: the vast majority of training data are held by the clients rather than the server.

**1. Model Output Comparison at Each Class:** When potential malicious clients are shortlisted, the server evaluates the output, which is typically a loss function, of each client in each class in $\mathcal{D}'$, i.e., $F(\boldsymbol{w}_{i,t}^K, \mathcal{D}'_c)$ ($\forall i \in \mathcal{M}'_t$), where $\mathcal{D}'_c$ denotes the subset of the dataset $\mathcal{D}'$ corresponding to class $c \in \mathcal{C}$. The server then compares the output of the FL model $F(\boldsymbol{w}_{i,t}^K, \mathcal{D}'_c)$ with that of the aggregated model based on all clients in $\mathcal{B}'_t$, i.e., $F(\boldsymbol{w}_{\mathcal{B}'_t}, \mathcal{D}'_c)$. Specifically, for each client $i \in \mathcal{M}'_t$, the server divides all classes into two categories: those for which $F(\boldsymbol{w}_{i,t}^K, \mathcal{D}'_c) < F(\boldsymbol{w}_{\mathcal{B}'_t}, \mathcal{D}'_c)$ and those for which $F(\boldsymbol{w}_{i,t}^K, \mathcal{D}'_c) \geq F(\boldsymbol{w}_{\mathcal{B}'_t}, \mathcal{D}'_c)$.

Let $\mathcal{C}_{1,i,t}$ and $\mathcal{C}_{2,i,t}$ denote the sets of classes for which $F(\boldsymbol{w}_{i,t}^K, \mathcal{D}'_c) < F(\boldsymbol{w}_{\mathcal{B}'_t}, \mathcal{D}'_c)$ and $F(\boldsymbol{w}_{i,t}^K, \mathcal{D}'_c) > F(\boldsymbol{w}_{\mathcal{B}'_t}, \mathcal{D}'_c)$, respectively. If $\mathcal{C}_{1,i,t}$ is empty, that is, $F(\boldsymbol{w}_{i,t}^K, \mathcal{D}'_c) > F(\boldsymbol{w}_{\mathcal{B}'_t}, \mathcal{D}'_c)$ in every class $c$, then the client is considered malicious. Otherwise, the server performs a statistical significance test on $F(\boldsymbol{w}_{i,t}^K, \mathcal{D}'_c)$ for all $c \in \mathcal{C}_{1,i,t}$.

**2. Significance Difference Test for Model Output at Each Class:** For each class in $\mathcal{C}_{1,i,t}$, the server performs a significance difference test using *Student's t-distribution*, comparing $F(\boldsymbol{w}_{i,t}^K, \mathcal{D}'_c)$ for $c \in \mathcal{C}_{1,i,t}$ with $F(\boldsymbol{w}_{i,t}^K, \mathcal{D}'_c)$ for all $c \in \mathcal{C}_{2,i,t}$. If the $p$-value of the significance test is lower than the threshold for every class $c \in \mathcal{C}_{1,i,t}$, then the client $i$ is considered to have rare data classes $c \in \mathcal{C}_{1,i,t}$. Otherwise, the client is considered malicious. Let $\mathcal{M}_t$ and $\mathcal{B}_t$ denote the sets of malicious and benign clients determined by the server, respectively, such that $\mathcal{M}_t \cup \mathcal{B}_t = \mathcal{N}_t$ and $\mathcal{M}_t \cap \mathcal{B}_t = \emptyset$. Note that $\mathcal{B}_t$ consists of all clients in $\mathcal{B}'_t$ as well as those in $\mathcal{M}'_t$ who passed the significance difference test.

Note that the significance difference test aims at discriminating benign and non-IID clients from malicious ones, as the server may identify the benign and non-IID clients as suspicious during the cosine-similarity-based filter.

### 3.3 Weight Updating Process

We propose a Bayesian dynamic reputation system that has the capacity to mitigate the presence of malicious clients and accounts for false alarms. This system is predicated on the modeling of each client's reputation as the probability of it being benign, based on its accumulated detection history. For a client $i$ at epoch $t$, the reputation score $r_{i,t}$ is the expected value of a Beta distribution:

$$r_{i,t} = \frac{\alpha_{i,t}}{\alpha_{i,t} + \beta_{i,t}}, \tag{6}$$

where $\alpha_{i,t}$ and $\beta_{i,t}$ serve as discounted evidence counters for benign and malicious behavior, respectively. These counters are updated at each epoch based on new detection results:

$$\alpha_{i,t} = \epsilon\alpha_{i,t-1} + \gamma_{i,t}, \ \beta_{i,t} = \epsilon\beta_{i,t-1} + 1 - \gamma_{i,t}. \tag{7}$$

Here, $\gamma_{i,t} = 1$ if client $i$ is detected as benign($i \in \mathcal{B}_t$) and $\gamma_{i,t} = 0$ otherwise. The discount factor $0 < \epsilon \leq 1$ prioritizes recent behavior, allowing reputations to adapt over time.

This reputation system is utilized in two distinct methods to secure the federated learning process:

**1. Robust Aggregation Weighting:** The final global model is aggregated using weights that are scaled by both the client's dataset size and its reputation. This measure is designed to guarantee that even in the event that a malevolent client manages to evade detection in a given round, its influence will invariably be suppressed by its comparatively low historical reputation.

$$\boldsymbol{w}_t = \sum_{i \in \mathcal{N}_t} \frac{r'_{i,t}|\mathcal{D}_i|}{\sum_{i \in \mathcal{N}_t} r'_{i,t}|\mathcal{D}_i|} \boldsymbol{w}_{i,t}^K, \tag{8}$$

where the effective reputation $r'_{i,t}$ provides a hard filter: we set $r'_{i,t} = r_{i,t}$ if $i \in \mathcal{B}_t$ but $r'_{i,t} = 0$ for any client flagged as malicious in the current round.

**2. Long-Term Client Selection:** The reputation score also functions as a long-term gatekeeping mechanism. Following an initial warm-up phase of $T_1$ epochs, only clients that have demonstrated consistent maintenance of a high reputation ($r_i > \underline{r}$) are considered eligible for subsequent training rounds.

The algorithmic details are delineated in the Appendix A.

Although `TriGuardFL` incorporates ideas appearing individually in prior Byzantine-robust FL works, its contribution lies in how the three stages are coordinated to distinguish benign non-IID divergence from malicious manipulation, which existing defenses do not explicitly target.

- Step 1 vs. `FLTrust`: `FLTrust` relies on a trusted model trained on a full clean dataset. `TriGuardFL` uses lightweight parameter-difference signals as a coarse filter rather than as the main decision rule.

- Step 2 vs. `DeFL`: `DeFL` conducts single-batch gradient-norm checks. `TriGuardFL` adopts class-wise, multi-batch discrepancy evaluation, which produces distinct signatures for benign skew versus adversarial optimization.

- Step 3 vs. generic reputation systems: The Bayesian update is tightly coupled to Step 2 and employs two-level gatekeeping (round-level hard filtering + gradual long-term weighting) to stabilize decisions under heterogeneity.

## 4 PERFORMANCE ANALYSIS

### 4.1 CONVERGENCE ANALYSIS

In this section, we derive the convergence analysis of `FedAvg` with our proposed `TriGuardFL` algorithm. Our convergence analysis refers to Li et al. (2019). In each epoch, malicious clients can bypass the detection of `TriGuardFL` and participate in parameter aggregation, as malicious clients may pass significant difference tests. Consequently, the poisoned stochastic gradients of the malicious clients have limited negative impact:

**Assumption 4.1.** In each malicious client that bypasses the proposed detection method, its variance of false stochastic gradients is bounded: $\mathbb{E}\|\nabla F_i'(\boldsymbol{w}_{i,t}) - \nabla F_i(\boldsymbol{w}_{i,t})\|^2 \leq \tau^2$, $t = 1, 2, ..., T$, where $F_i'(\boldsymbol{w}_{i,t})$ denotes the false gradients.

Let $\mathcal{M}$ and $\mathcal{B}$ denote the sets of malicious and benign clients, respectively, and let $\mathcal{M} \cup \mathcal{B} = \mathcal{N}$ and $\mathcal{M} \cap \mathcal{B} = \emptyset$ hold. We consider the non-IID degree of both benign and malicious clients, and we have the following theorem.

**Theorem 4.2.** Let Assumptions 2.1-4.1 hold and $L$, $\mu$, $\sigma_i$, $\tau$, and $G$ be defined therein. Set $\kappa = \frac{L}{\mu}$, $\gamma = \max\{8\kappa, 1\}$ and the learning rate $\eta_{i,t} = \frac{2}{\mu(\gamma + K(t-1))}$. Then `FedAvg` with selected device participation satisfies

$$\mathbb{E}[F(\boldsymbol{w}_{T+1})] - F^* \leq \frac{\kappa}{\gamma + KT}\left(\frac{2(B+C)}{\mu} + \frac{\mu\gamma}{2}\mathbb{E}\|\boldsymbol{w}_1 - \boldsymbol{w}^*\|_2^2\right) + \epsilon(\mathcal{M}), \tag{9}$$

where

$$B = 6L\Gamma + 8(K-1)^2 G^2, \ C = 4K^2 G^2, \ \epsilon(\mathcal{M}) = \sum_{i \in \mathcal{M}} \frac{|\mathcal{D}_i|}{|\mathcal{D}|} \frac{8(2(K-1)^2 + K^2)\kappa\tau^2}{\mu(\gamma + KT)}, \tag{10}$$

and $\epsilon(\mathcal{M})$ denotes the impact of malicious clients that involves aggregation of model parameters.

*Proof.* See Appendix C. $\square$

Theorem 4.2 directly considers the general case: At each epoch, some of the selected clients are benign, while others are malicious. Therefore, (9) indicates the lower bound of the convergence rate. In practice, if malicious clients are detected in part of the epochs or removed from the client selection by `TriGuardFL`, the global FL model converges faster.

## 4.2 ROBUSTNESS ANALYSIS

`TriGuardFL` is robust to model poisoning attacks, while keeps benign clients with non-IID data, because the latter step performs the fault toleration for failure of the previous step in the three-step detection and protection mechanism. Specifically, the first and third steps detect malicious clients, while the second step recognizes benign clients who possess a unique data sample. Especially, the third step is robust to malicious clients. Even if malicious clients bypass the cosine-similarity-based detection or the statistical significance test at most epochs, the server rapidly removes them from the model aggregation list if they are considered as malicious for several times. For clients who possess unique data sample, the server may shortlist the clients to the set of potential malicious clients during the cosine-similarity-based detection, but the clients can pass the statistical significance test in the second step and be recognized as benign.

The following performance on `TriGuardFL` is achieved.

**Theorem 4.3.** *Based on Assumptions 2.1-4.1, when all malicious clients are removed from the long-term client selection list at epoch $t \geq T_1$, the FL model parameter satisfies*

$$\mathbb{E}\|\boldsymbol{w}_t - \boldsymbol{w}^*\|^2 \leq (1 - \eta\mu)^{t-1} \mathbb{E}\|\boldsymbol{w}_1 - \boldsymbol{w}^*\|^2 + (B + C)/\mu. \tag{11}$$

*When $\eta\mu \leq 1$, we have $\lim_{t \to \infty} \mathbb{E}\|\boldsymbol{w}_t - \boldsymbol{w}^*\|^2 = (B + C)/\mu$.*

*Proof.* See Appendix D. $\square$

## 5 EXPERIMENTS

In this section, we evaluate the proposed `TriGuardFL` algorithm and compare it with other state-of-the-art defense algorithms against model poisoning attacks in a heterogeneous, non-IID setting. We aim to determine the algorithm's effectiveness against established defense mechanisms across different deep learning models and its resilience under escalating threat levels, especially as the proportion of malicious clients in the network increases.

### 5.1 EXPERIMENTAL SETUP

#### 5.1.1 DATASETS AND MACHINE LEARNING ARCHITECTURE

Our evaluation uses two widely adopted benchmark datasets: **Fashion-MNIST** (Xiao et al., 2017) and **CIFAR-10** (Krizhevsky, 2009). To emulate a realistic federated environment with heterogeneous data, we partition samples among clients using a Dirichlet distribution ($\boldsymbol{p} \sim \text{Dir}_N(\alpha)$). We set the concentration parameter to $\alpha = 0.5$, a standard choice in the literature that induces a significant degree of non-IID distribution in the local datasets of the clients.

The experiments employ five representative neural network architectures to assess performance across varying model complexities: LeNet (LeCun et al., 2002), AlexNet (Krizhevsky et al., 2012), VGG11 (Simonyan & Zisserman, 2015), VGG16 (Simonyan & Zisserman, 2015), and ResNet18 (He et al., 2016). Specifically, we pair the LeNet model with the Fashion-MNIST dataset. For the more complex CIFAR-10 dataset, we evaluate performance using AlexNet, VGG11, VGG16, and ResNet18 to cover a diverse set of architectures.

#### 5.1.2 BASELINE DEFENSES AND ATTACKS

To benchmark the performance of `TriGuardFL`, we compare it against four methods: DeFL, FL-WBC, FLTrust, and Multi-Krum. With regard to the attack models, three model poisoning attacks are adopted: Min-SumShejwalkar & Houmansadr (2021), Min-Max Shejwalkar & Houmansadr (2021), and LIE Baruch et al. (2019). Following standard practice in the literature Baruch et al. (2019); Fang et al. (2020); Shejwalkar & Houmansadr (2021); Yan et al. (2023), we assess performance under two threat models, which are defined by the adversary's level of knowledge:

- **Full Knowledge:** The malicious clients have complete knowledge of the gradients of benign clients in the current round, allowing them to craft an optimal attack.

- **Partial Knowledge:** Malicious clients have no knowledge of the gradients of benign clients and must build their attack based solely on the previous global model.

The key hyperparameters used throughout the experiments are summarized in Table 1.

Table 1: Default hyperparameter configuration.

| Parameter | Total Clients ($N$) | Malicious Clients ($m$) | Clients per Round | Local Iterations ($K$) | Learning Rate ($\eta$) | Cosine-Similarity Threshold ($\delta_1$) | Significance Level | Reputation Threshold ($\underline{r}$) |
|---|---|---|---|---|---|---|---|---|
| **Value** | 32 | 4 (12.5%) | 16 | 4 | 0.05 | 0.1 | 0.001 | 0.6 |

## 5.2 Experimental Results

### 5.2.1 Mitigating the impact of attacks

We subjected the FL system to three potent attacks: Min-Max, Min-Sum, and LIE under partial- and full-knowledge scenarios.

Table 2 presents the final test loss for all evaluated defenses across the five network architectures. According to the average performance rank, `TriGuardFL` is the highest-performing defense, achieving rank 1 in three of the five experimental settings (LeNet, AlexNet, VGG-16) and tied for first in a fourth (ResNet-18). This resilience is particularly evident under strong, full-knowledge attacks. For example, when defending a ResNet-18 model against the Min-Max attack, TriGuardFL maintained a test loss of 1.367; meanwhile, the losses for Multi-Krum and DeFL significantly degraded to 4.875 and 4.084, respectively. Note that the losses in some "Partial" cases are higher than that in "Full" cases, as the attack strategy is more aggressive in these "Partial" cases.

| Dataset (Model) | Byzantine-robust Rules | Min-Max | | Min-Sum | | LIE | | Rank |
|---|---|---|---|---|---|---|---|---|
| | | Partial | Full | Partial | Full | Partial | Full | |
| Fashion MNIST (LeNet) | TriGuardFL | 0.393±0.050 | 0.371±0.039 | 0.350±0.053 | 0.393±0.059 | 0.390±0.033 | 0.363±0.041 | 1 |
| | DeFL | 0.369±0.047 | 0.431±0.114 | 0.391±0.031 | 0.413±0.051 | 0.374±0.053 | 0.376±0.056 | 2 |
| | FLTrust | 0.475±0.084 | 0.445±0.099 | 0.401±0.076 | 0.419±0.087 | 0.412±0.054 | 0.393±0.058 | 5 |
| | FLDetector | 0.431±0.064 | 0.442±0.041 | 0.392±0.067 | 0.791±0.198 | 0.389±0.060 | 0.357±0.052 | 3 |
| | Multi-Krum | 0.419±0.046 | 0.505±0.102 | 0.538±0.139 | 0.378±0.031 | 0.367±0.030 | 0.392±0.044 | 4 |
| | No attack | | | 0.346±0.040 | | | | 0 |
| CIFAR-10 (AlexNet) | TriGuardFL | 1.975±0.187 | 1.824±0.217 | 1.937±0.201 | 1.685±0.268 | 1.883±0.211 | 1.897±0.105 | 1 |
| | DeFL | 2.401±0.382 | 1.883±0.217 | 2.299±0.273 | 1.807±0.122 | 1.972±0.134 | 1.964±0.196 | 4 |
| | FLTrust | 1.792±0.192 | 1.854±0.119 | 1.983±0.290 | 1.891±0.132 | 1.967±0.154 | 1.916±0.241 | 3 |
| | FLDetector | 1.860±0.140 | 1.756±0.080 | 2.233±0.822 | 2.048±0.345 | 1.722±0.195 | 1.778±0.119 | 2 |
| | Multi-Krum | 2.642±0.398 | 2.493±0.277 | 2.398±0.320 | 1.936±0.093 | 1.836±0.163 | 1.920±0.283 | 5 |
| | No attack | | | 1.561±0.154 | | | | 0 |
| CIFAR-10 (VGG-11) | TriGuardFL | 1.494±0.363 | 1.194±0.111 | 1.219±0.087 | 1.329±0.078 | 1.239±0.101 | 1.145±0.050 | 2 |
| | DeFL | 1.654±0.280 | 2.173±0.906 | 1.583±0.354 | 1.682±0.393 | 1.375±0.214 | 1.349±0.148 | 5 |
| | FLTrust | 1.143±0.241 | 1.443±0.104 | 1.169±0.234 | 1.215±0.164 | 1.265±0.157 | 1.348±0.176 | 3 |
| | FLDetector | 1.381±0.128 | 1.340±0.104 | 1.198±0.158 | 1.349±0.112 | 1.213±0.135 | 1.134±0.076 | 1 |
| | Multi-Krum | 2.393±0.807 | 2.141±0.732 | 2.088±1.388 | 3.371±0.225 | 1.225±0.041 | 1.310±0.160 | 4 |
| | No attack | | | 1.061±0.164 | | | | 0 |
| CIFAR-10 (VGG-16) | TriGuardFL | 1.333±0.100 | 1.240±0.215 | 1.279±0.085 | 1.257±0.101 | 1.169±0.160 | 1.078±0.130 | 1 |
| | DeFL | 1.422±0.159 | 1.739±0.244 | 1.751±0.245 | 1.662±0.661 | 1.277±0.138 | 1.321±0.187 | 4 |
| | FLTrust | 1.331±0.155 | 1.304±0.080 | 1.231±0.210 | 1.622±0.229 | 1.181±0.096 | 1.329±0.110 | 2 |
| | FLDetector | 1.545±0.301 | 1.468±0.157 | 1.361±0.414 | 1.537±0.255 | 1.183±0.236 | 1.212±0.072 | 3 |
| | Multi-Krum | 2.170±0.415 | 4.139±1.195 | 1.878±0.383 | 1.787±0.408 | 1.206±0.218 | 1.128±0.118 | 5 |
| | No attack | | | 1.122±0.037 | | | | 0 |
| CIFAR-10 (ResNet-18) | TriGuardFL | 1.585±0.228 | 1.367±0.091 | 1.912±0.765 | 1.324±0.129 | 1.352±0.148 | 1.391±0.109 | 1 |
| | DeFL | 2.376±0.819 | 4.084±1.217 | 1.983±0.332 | 1.936±0.217 | 1.560±0.196 | 1.526±0.056 | 5 |
| | FLTrust | 1.410±0.061 | 1.466±0.106 | 1.308±0.207 | 1.429±0.259 | 1.353±0.088 | 1.365±0.101 | 1 |
| | FLDetector | 1.587±0.261 | 1.747±0.195 | 1.443±0.146 | 2.013±0.238 | 1.250±0.126 | 1.227±0.110 | 3 |
| | Multi-Krum | 2.193±0.314 | 4.875±1.705 | 1.981±0.449 | 3.063±1.152 | 1.375±0.146 | 1.305±0.129 | 4 |
| | No attack | | | 1.172±0.116 | | | | 0 |

Table 2: Loss of two test datasets with five categories of neural networks under three state-of-the-art model poisoning attacks defended by `TriGuardFL` and four other defense algorithms when the training dataset is non-IID.

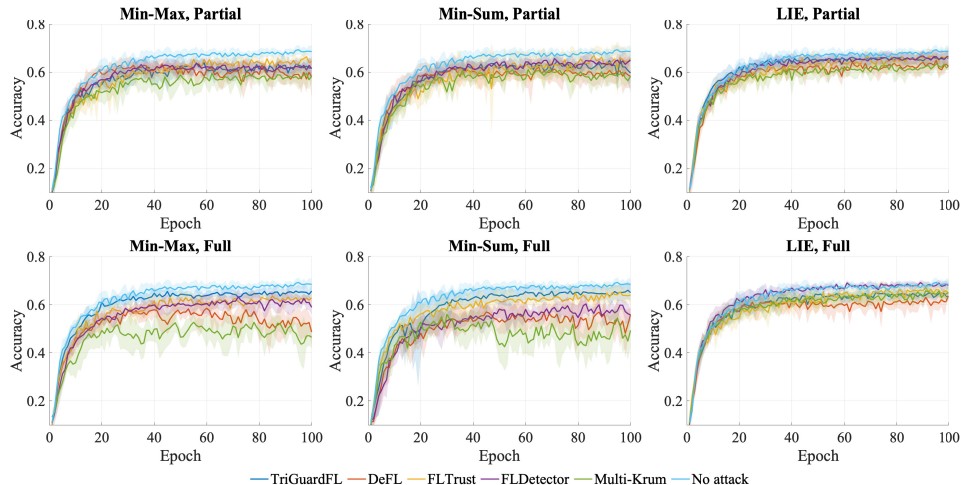

Figure 1: Evolution of test accuracy on CIFAR-10, comparing `TriGuardFL` against baseline defenses under various model poisoning attacks. The experimental setup uses a ResNet18 model trained on non-IID data distributed among 32 total clients, including 4 malicious attackers.

Figure 1 shows that the accuracy curve for `TriGuardFL` remains high and stable throughout the training process, closely tracking the performance of the ideal "no attack" baseline. This stands in contrast to several baseline defenses, such as DeFL and Multi-Krum, which exhibit significant performance degradation and instability, particularly under the more sophisticated full-knowledge attacks. These results underscore `TriGuardFL`'s ability to effectively filter out malicious updates without incorrectly penalizing benign clients, which is a critical weakness in many defenses that struggle with non-IID data.

### 5.2.2 ROBUSTNESS AGAINST AN INCREASING NUMBER OF ADVERSARIES

To assess the scalability of our method, we conducted experiments in more hostile environments by increasing the network scale and the proportional number of adversaries. This analysis investigates how defense mechanisms perform under heightened threat levels, a critical consideration for real-world deployment.

Table 3 compares the test loss of the defense mechanisms in a baseline configuration and a scaled network configuration with a proportionally larger number of attackers. The results highlight the superior scalability of `TriGuardFL`. As the threat level intensifies, `TriGuardFL` is the only defense that maintains its top-ranked performance. In contrast, the effectiveness of other defenses degrades sharply; for instance, the test loss for DeFL and Multi-Krum increases dramatically, indicating a performance collapse. This demonstrates that `TriGuardFL`'s detection and reputation mechanisms are robust to scaling, making it a more reliable defense for larger real-world FL systems.

More experimental results are shown in the Appendix B.

## 6 CONCLUSION

In this paper, we propose a novel Byzantine robust FL algorithm named `TriGuardFL`. Unlike previous approaches that compare the parameters or gradients of different clients, which are limited in the non-IID conditions, `TriGuardFL` detects malicious clients by analyzing the output of the objective function via significance difference test. To enhance the robustness of `TriGuardFL`, a Bayesian-based reputation model is utilized to eliminate the uncertainty of detection. Hence, `TriGuardFL` has excellent character in terms of the non-IID dataset distribution. The results demonstrate that `TriGuardFL` achieves better performances than baseline defense rules in terms of model accuracy and robustness. However, `TriGuardFL` rely on an independent dataset on the server, which restricts the application scenarios. In the future, we will explore defense algorithms for FL that do not rely on the server's independent dataset.

| Number of Clients/ Chosen Clients/ Attackers | Byzantine-robust Rules | Attacks | | | | | | Rank |
|---|---|---|---|---|---|---|---|---|
| | | Min-Max | | Min-Sum | | LIE | | |
| | | Partial | Full | Partial | Full | Partial | Full | |
| | `TriGuardFL` | 1.585±0.228 | 1.367±0.091 | 1.912±0.765 | 1.324±0.129 | 1.352±0.148 | 1.391±0.109 | 1 |
| | DeFL | 2.376±0.819 | 4.084±1.217 | 1.983±0.332 | 1.936±0.217 | 1.560±0.196 | 1.526±0.056 | 5 |
| 32/16/4 | FLTrust | 1.410±0.061 | 1.466±0.106 | 1.308±0.207 | 1.429±0.259 | 1.353±0.088 | 1.365±0.101 | 1 |
| | FLDetector | 1.587±0.261 | 1.747±0.195 | 1.443±0.146 | 2.013±0.238 | 1.250±0.126 | 1.227±0.110 | 3 |
| | Multi-Krum | 2.193±0.314 | 4.875±1.705 | 1.981±0.449 | 3.063±1.152 | 1.375±0.146 | 1.305±0.129 | 4 |
| | No attack | 1.172±0.116 | | | | | | 0 |
| | `TriGuardFL` | 1.733±0.060 | 1.427±0.043 | 2.712±0.542 | 1.433±0.071 | 1.404±0.101 | 1.415±0.090 | 1 |
| | DeFL | 4.927±2.480 | 2.653±0.661 | 3.305±3.644 | 1.447±0.073 | 1.624±0.079 | 1.766±0.132 | 5 |
| 64/32/8 | FLTrust | 1.839±0.229 | 1.696±0.266 | 1.756±0.220 | 1.732±0.129 | 1.556±0.129 | 1.669±0.270 | 2 |
| | FLDetector | 1.918±0.151 | 1.813±0.135 | 2.886±0.609 | 3.418±0.202 | 1.386±0.081 | 1.439±0.116 | 2 |
| | Multi-Krum | 3.590±0.407 | 8.975±7.350 | 11.847±5.791 | 3.052±2.857 | 1.483±0.073 | 1.521±0.139 | 4 |
| | No attack | 1.350±0.099 | | | | | | 0 |

Table 3: Loss of test dataset CIFAR-10 under three state-of-the-art model poisoning attacks with different clients and attackers defended by `TriGuardFL` and another four defense algorithms when the training dataset is non-IID and the neural network is ResNet18.

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

## A PSEUDOCODE FOR TRIGUARDFL

---

**Algorithm 1** The `TriGuardFL` algorithm

---

**Input**: $\forall i, \boldsymbol{w}_i(0), \mathcal{D}', \delta_1, \delta_2, \underline{r}$;
**Initialize**: $\forall i, \alpha_{i,0} = \beta_{i,0} = 1$;

1:  **for** $t = 1, 2, ..., T$ **do**
2:      **if** $t < T_1$ **then**
3:          Ramdomly select several clients from all clients in $\mathcal{N}$ for parameter update;
4:      **else**
5:          Ramdomly select several clients from clients with $r_{i,t} > \underline{r}$ for parameter update;
6:      **end if**
7:      Aggregate the parameter of selected clients and obtains $\boldsymbol{w}'_t$ by (3);
8:      Calculate the cosine similarity of each client $\cos\theta_{i,t}$ by (4);
9:      **for** $i \in \mathcal{N}_t$ **do**
10:          **if** $\cos\theta_i < \delta_1$ **then**
11:              $\mathcal{M}'_t \leftarrow \mathcal{M}'_t \cup \{i\}$;
12:          **else**
13:              $\mathcal{B}'_t \leftarrow \mathcal{B}'_t \cup \{i\}$;
14:          **end if**
15:      **end for**
16:      Aggregate the parameter of all clients in $\mathcal{B}'_t$ and obtains $\boldsymbol{w}_{\mathcal{B}'_t}$ by (3);
17:      Compute $F(\boldsymbol{w}_{\mathcal{B}'_t}, \mathcal{D}'_c)$ for each class $c$;
18:      **for** $i \in \mathcal{M}'_t$ **do**
19:          Compute $F(\boldsymbol{w}^K_{i,t}, \mathcal{D}'_c)$ for each class $c$;
20:          **for** $c \in \mathcal{C}$ **do**
21:              **if** $F(\boldsymbol{w}^K_{i,t}, \mathcal{D}'_c) < F(\boldsymbol{w}_{\mathcal{B}'_t}, \mathcal{D}'_c)$ **then**
22:                  $\mathcal{C}_{1,i,t} \leftarrow \mathcal{C}_{1,i,t} \cup \{c\}$;
23:              **else**
24:                  $\mathcal{C}_{2,i,t} \leftarrow \mathcal{C}_{2,i,t} \cup \{c\}$;
25:              **end if**
26:          **end for**
27:          **for** $c \in \mathcal{C}_{1,i,t}$ **do**
28:              Compare $F(\boldsymbol{w}^K_{i,t}, \mathcal{D}'_c)$ with $\{F(\boldsymbol{w}^K_{i,t}, \mathcal{D}'_c)\}_{c \in \mathcal{C}_{2,i,t}}$ by t-test for statistical significance;
29:          **end for**
30:          **if** $p < \delta_2$ for each class $c \in \mathcal{C}_{1,i,t}$ **then**
31:              $\mathcal{B}_t \leftarrow \mathcal{B}_t \cup \{i\}$;
32:          **else**
33:              $\mathcal{M}_t \leftarrow \mathcal{M}_t \cup \{i\}$;
34:          **end if**
35:      **end for**
36:      $\mathcal{B}_t \leftarrow \mathcal{B}_t \cup \mathcal{B}'_t$;
37:      **for** $i \in \mathcal{N}_t$ **do**
38:          $\gamma_t = 1$ if $i \in \mathcal{B}_t$, $\gamma_t = 0$ otherwise;
39:          Update $\alpha_{i,t}$ and $\beta_{i,t}$ by (7);
40:          Update $r_{i,t}$ by (6);
41:      **end for**
42:      The aggregation weight for the client $i \in \mathcal{M}_t$ (resp. $i \in \mathcal{B}_t$) is $r'_{i,t} = 0$ (resp. (6));
43:      Aggregate the global weight $\boldsymbol{w}_t$ via `FedAvg` (8).
44: **end for**

---

## B ADDITIONAL EXPERIMENTAL RESULTS

### B.1 ACCURACY

Figure 2-4 depict the test accuracy under three state-of-the-art model poisoning attacks, defended by `TriGuardFL` and four other defense algorithms when the training dataset is non-IID with Dirichlet

distribution $\alpha = 0.5$. Notably, `TriGuardFL` consistently exhibits the highest accuracy in most cases, indicating its robustness against model poisoning attacks.

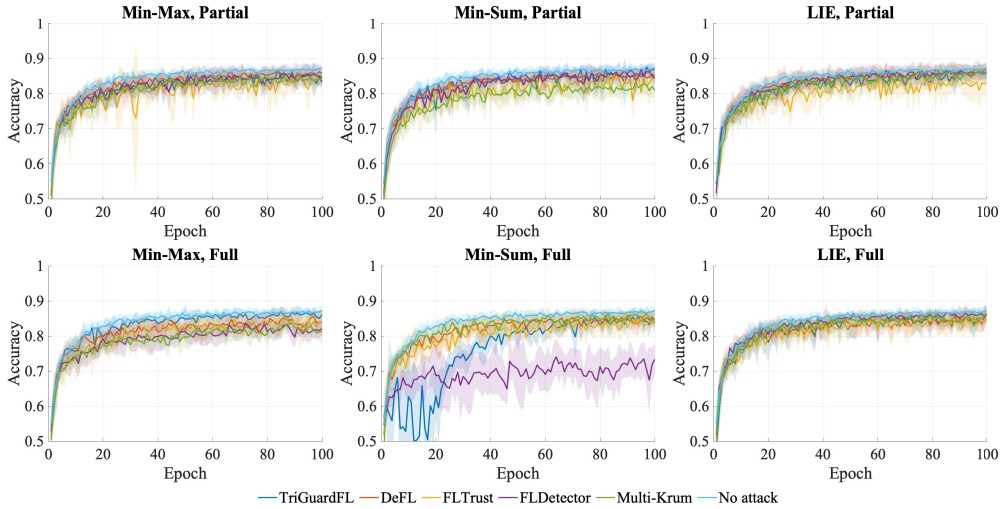

Figure 2: Accuracy of test dataset Fashion-MNIST under three state-of-the-art model poisoning attacks defended by `TriGuardFL` and another four defense algorithms when the training dataset is non-IID and the neural network model is LeNet.

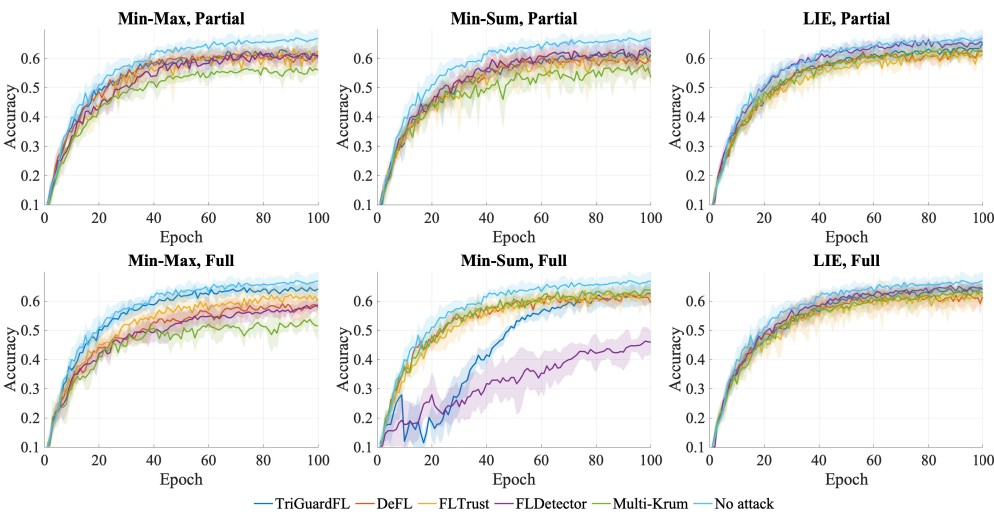

Figure 3: Accuracy of test dataset CIFAR-10 under three state-of-the-art model poisoning attacks defended by `TriGuardFL` and another four defense algorithms when the training dataset is non-IID and the neural network model is AlexNet.

### B.2 IMPACT OF THE NUMBER OF ADVERSARIES.

The number of malicious clients significantly influences the accuracy of the FL model accuracy. In our previous experiments, we set 4 malicious clients as default. To evaluate the detection ability and model accuracy of `TriGuardFL` under different numbers of adversaries, we conducted additional tests with 5 and 6 malicious clients based on the CIFR-10 dataset and the ResNet18 neural network model, while keeping other settings constant. As shown in Table 4 and Figure 6, the accuracy of the test dataset decreases and the losses of the test dataset increase in general. For `TriGuardFL`, the losses of the test dataset do not have significant differences with different numbers of malicious clients, indicating that `TriGuardFL` keeps excellent robustness. Moreover, DeFL and Multi-Krum

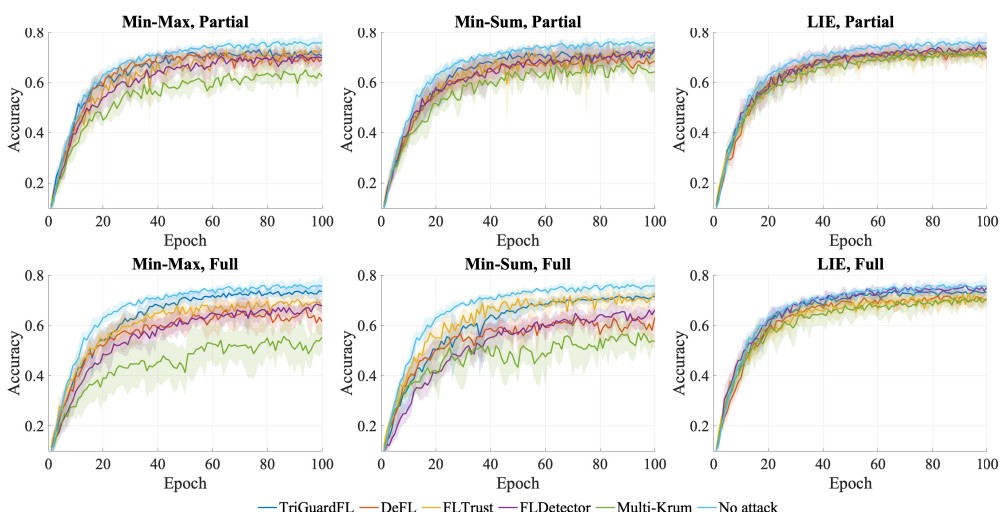

Figure 4: Accuracy of test dataset CIFAR-10 under three state-of-the-art model poisoning attacks defended by `TriGuardFL` and another four defense algorithms when the training dataset is non-IID and the neural network model is VGG11.

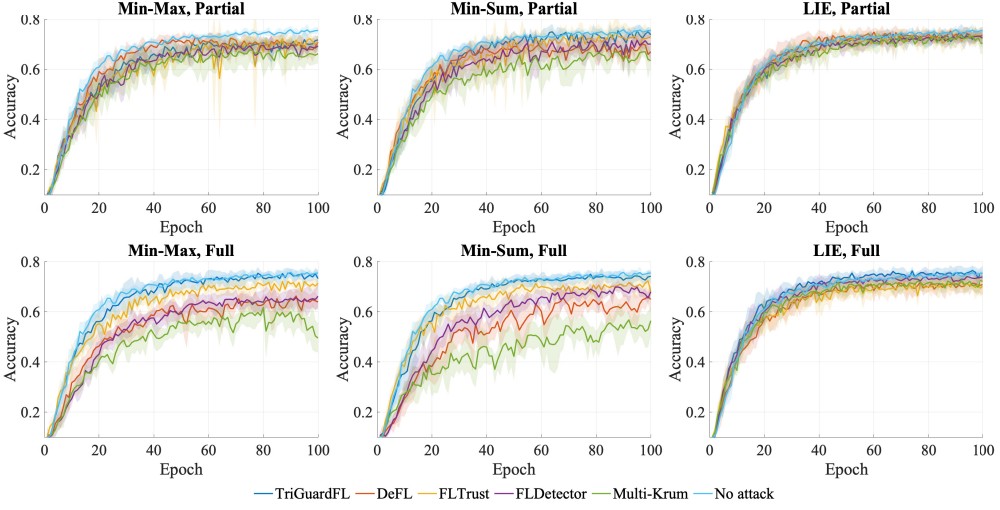

Figure 5: Accuracy of test dataset CIFAR-10 under three state-of-the-art model poisoning attacks defended by `TriGuardFL` and another four defense algorithms when the training dataset is non-IID and the neural network model is VGG16.

crash in part of the cases. This is because even if `TriGuardFL` can detect most malicious clients, the number of benign clients decreases as that of malicious clients increases, and samples in benign clients also decrease.

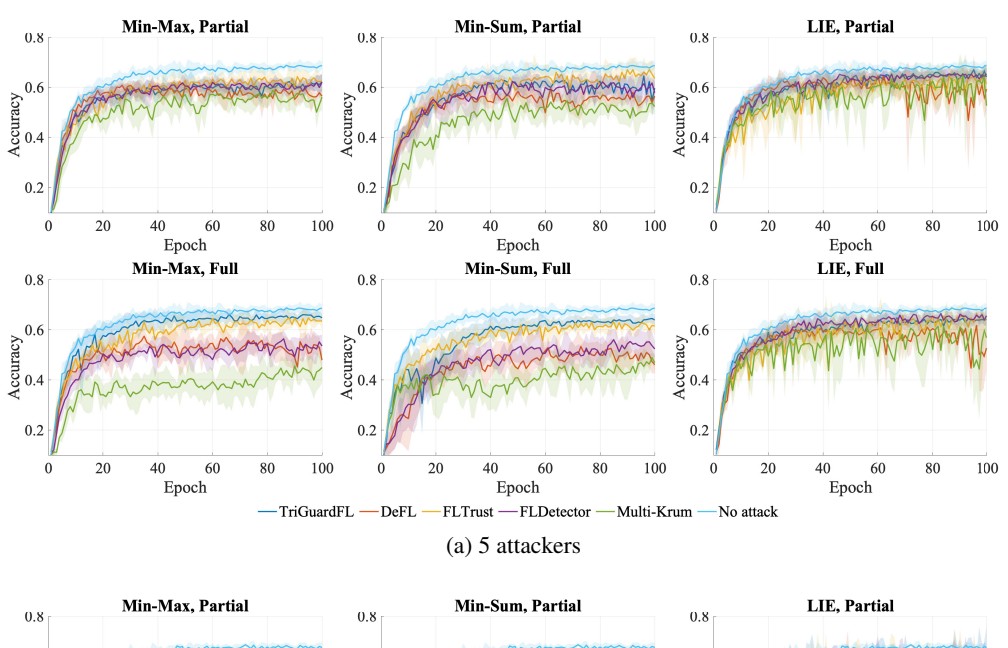

(a) 5 attackers

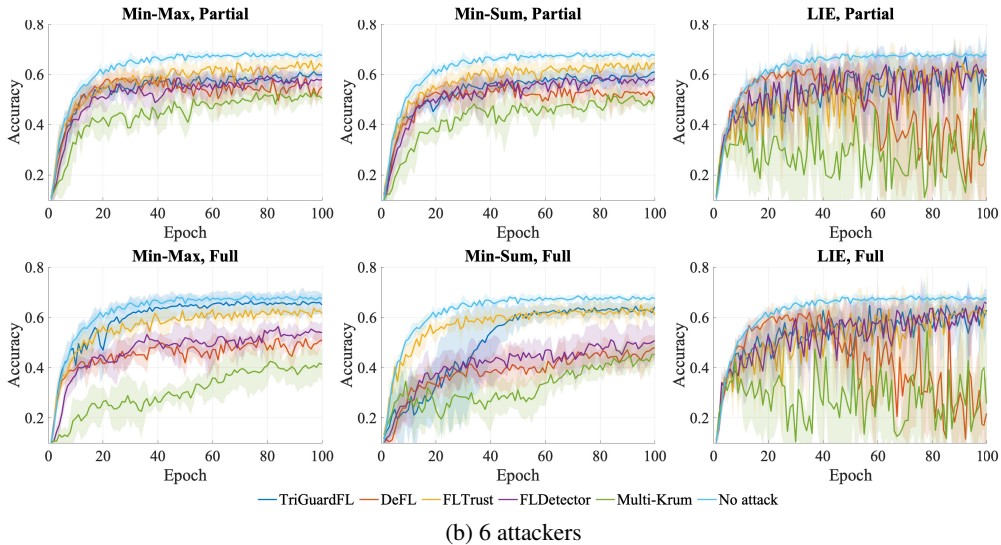

(b) 6 attackers

Figure 6: Accuracy of test dataset CIFAR-10 under three state-of-the-art model poisoning attacks with more attackers defended by `TriGuardFL` and another four defense algorithms when the training dataset is non-IID and the neural network model is ResNet18.

### B.3 IMPACT OF THE NUMBER OF TOTAL CLIENTS.

Figure 7 illustrates the evolution of the test accuracy of `TriGuardFL` with another four other state-of-the-art defense rules based on the CIFR-10 dataset and the ResNet18 neural network model when the number of clients is 64 and that of attackers is 8. Compared with Figure 1, the accuracy of five defense rules decreases in general. It is obvious that `TriGuardFL` and FLTrust achieve near convergence, while DeFL and Multi-Krum crash in some cases.

| Number of Attackers | Byzantine-robust Rules | Attacks | | | | | | Rank |
|---|---|---|---|---|---|---|---|---|
| | | Min-Max | | Min-Sum | | LIE | | |
| | | Partial | Full | Partial | Full | Partial | Full | |
| 4 | TriGuardFL | 1.585±0.228 | 1.367±0.091 | 1.912±0.765 | 1.324±0.129 | 1.352±0.148 | 1.391±0.109 | 1 |
| | DeFL | 2.376±0.819 | 4.084±1.217 | 1.983±0.332 | 1.936±0.217 | 1.560±0.196 | 1.526±0.056 | 5 |
| | FLTrust | 1.410±0.061 | 1.466±0.106 | 1.308±0.207 | 1.429±0.259 | 1.353±0.088 | 1.365±0.101 | 1 |
| | FLDetector | 1.587±0.261 | 1.747±0.195 | 1.443±0.146 | 2.013±0.238 | 1.250±0.126 | 1.227±0.110 | 3 |
| | Multi-Krum | 2.193±0.314 | 4.875±1.705 | 1.981±0.449 | 3.063±1.152 | 1.375±0.146 | 1.305±0.129 | 4 |
| | No attack | 1.172±0.116 | | | | | | 0 |
| 5 | TriGuardFL | 1.853±0.191 | 1.402±0.104 | 1.921±0.751 | 1.414±0.072 | 1.457±0.247 | 1.414±0.142 | 1 |
| | DeFL | 2.558±0.365 | 3.168±0.696 | 2.767±0.888 | 2.760±0.362 | 2.186±1.439 | 2.631±1.469 | 4 |
| | FLTrust | 1.504±0.134 | 1.421±0.137 | 1.469±0.220 | 1.532±0.193 | 1.595±0.301 | 1.454±0.331 | 2 |
| | FLDetector | 1.824±0.245 | 2.179±0.346 | 2.281±0.419 | 2.092±0.287 | 1.362±0.107 | 1.320±0.178 | 3 |
| | Multi-Krum | 2.844±0.595 | 7.486±5.936 | 2.941±1.011 | 6.863±3.941 | 3.588±3.728 | 2.238±1.304 | 5 |
| | No attack | 1.201±0.057 | | | | | | 0 |
| 6 | TriGuardFL | 2.166±0.132 | 1.401±0.222 | 1.781±0.433 | 1.322±0.089 | 3.809±5.651 | 1.612±0.563 | 2 |
| | DeFL | 3.496±1.044 | 2.791±0.600 | 4.019±1.275 | 7.020±9.552 | 120.780±187.637 | 79.853±86.309 | 4 |
| | FLTrust | 1.458±0.083 | 1.584±0.146 | 1.436±0.037 | 1.563±0.258 | 1.465±0.269 | 1.605±0.235 | 1 |
| | FLDetector | 2.538±0.293 | 2.201±0.449 | 2.295±0.378 | 2.212±0.377 | 2.008±0.762 | 1.338±0.020 | 3 |
| | Multi-Krum | 6.453±4.198 | 34.498±22.391 | 4.235±2.017 | 25.757±13.735 | 88.962±116.521 | 243.309±403.874 | 5 |
| | No attack | 1.234±0.080 | | | | | | 0 |

Table 4: Lose of test dataset CIFAR-10 under three state-of-the-art model poisoning attacks with 4/5/6 attackers defended by TriGuardFL and another four defense algorithms when the training dataset is non-IID and the neural network is ResNet18.

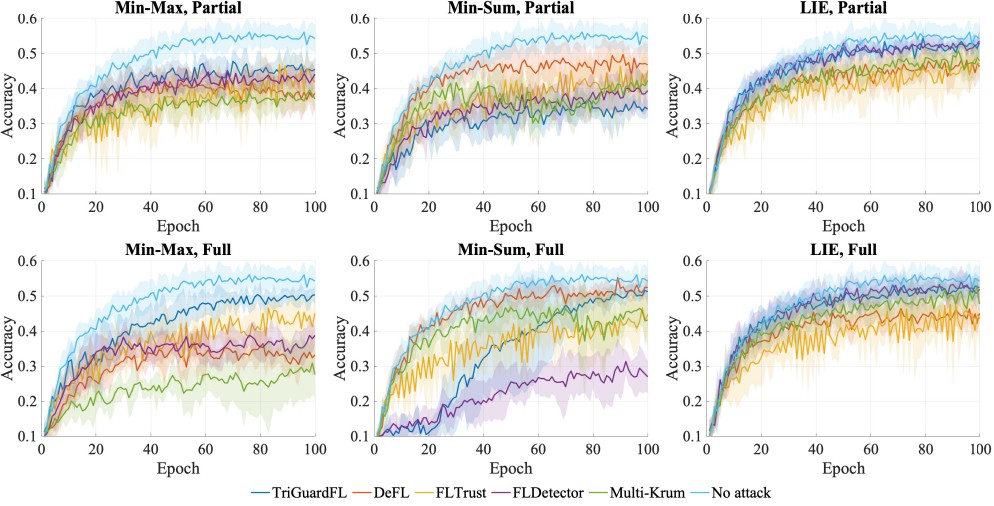

Figure 7: Accuracy of test dataset CIFAR-10 under three state-of-the-art model poisoning attacks defended by TriGuardFL and another four defense algorithms when the number of clients is 64 and that of attackers is 8.

## C    PROOF OF THEOREM 4.2

*Proof.* We prove the convergence rate in the existence of malicious clients. Inspired by Li et al. (2019); Sun et al. (2021), we first derive the expected upper bound of malicious clients, then we derive the convergence rate with the expected upper bound of malicious clients and benign clients.

VARIANCE OF FALSE STOCHASTIC GRADIENTS IN EACH MALICIOUS CLIENTS.

Considering Assumption 2.3 and 4.1, the expected false stochastic gradients of malicious clients is

$$
\mathbb{E}\|\nabla F_i'(\boldsymbol{w}_{i,t})\|^2
$$
$$
=\mathbb{E}\|\nabla F_i'(\boldsymbol{w}_{i,t}) - \nabla F_i(\boldsymbol{w}_{i,t}) + \nabla F_i(\boldsymbol{w}_{i,t})\|^2
$$
$$
\leq \mathbb{E}\|\nabla F_i'(\boldsymbol{w}_{i,t}) - \nabla F_i(\boldsymbol{w}_{i,t})\|^2 + \mathbb{E}\|\nabla F_i(\boldsymbol{w}_{i,t})\|^2
$$
$$
\leq \tau^2 + G^2. \tag{12}
$$
$$
\mathbb{E}\left[\|\sum_{i\in\mathcal{B}}\frac{|\mathcal{D}_i|}{|\mathcal{D}|}\nabla F_i(\boldsymbol{w}_{i,t}) + \sum_{i\in\mathcal{M}}\frac{|\mathcal{D}_i|}{|\mathcal{D}|}\nabla F_i'(\boldsymbol{w}_{i,t})\|^2\right]
$$
$$
\leq \sum_{i\in\mathcal{B}}\frac{|\mathcal{D}_i|}{|\mathcal{D}|}\mathbb{E}\|\nabla F_i(\boldsymbol{w}_{i,t})\|^2 + \sum_{i\in\mathcal{M}}\frac{|\mathcal{D}_i|}{|\mathcal{D}|}\mathbb{E}\|\nabla F_i'(\boldsymbol{w}_{i,t})\|^2
$$
$$
\leq \frac{\sum_{i\in\mathcal{M}}|\mathcal{D}_i|}{|\mathcal{D}|}\tau^2 + G^2. \tag{13}
$$

CONVERGENCE RATE OF FEDAVG IN THE PRESENCE OF MALICIOUS CLIENTS.

We assume that each client takes $K$ times of local training in each epoch. Let Assumptions 2.1-2.3 hold and $L$, $\mu$, $\sigma_i$, $\tau$, and $G$ be defined therein. Set $\kappa = \frac{L}{\mu}$, $\gamma = \max\{8\kappa, 1\}$ and the learning rate $\eta_{i,t} = \frac{2}{\mu(\gamma+K(t-1))}$. By referring Li et al. (2019), if all clients are benign, we have

$$
\mathbb{E}[F(\boldsymbol{w}_{T+1})] - F^* \leq \frac{\kappa}{\gamma + KT}\left(\frac{2(B+C)}{\mu} + \frac{\mu\gamma}{2}\mathbb{E}\|\boldsymbol{w}_1 - \boldsymbol{w}^*\|_2^2\right), \tag{14}
$$

where

$$
B = 6L\Gamma + 8(K-1)^2 G^2,
$$
$$
C = 4K^2 G^2.
$$

Then we consider the case that malicious clients involve the updating of parameters. By referring Lemma 2-5 in Li et al. (2019) and (13), we can obtain that

$$
B' = 6L\Gamma + 8(K-1)^2\left(\frac{\sum_{i\in\mathcal{M}}|\mathcal{D}_i|}{|\mathcal{D}|}\tau^2 + G^2\right) = B + 8\frac{\sum_{i\in\mathcal{M}}|\mathcal{D}_i|}{|\mathcal{D}|}(K-1)^2\tau^2,
$$
$$
C' = 4K^2\left(\frac{\sum_{i\in\mathcal{M}}|\mathcal{D}_i|}{|\mathcal{D}|}\tau^2 + G^2\right) = C + 4\frac{\sum_{i\in\mathcal{M}}|\mathcal{D}_i|}{|\mathcal{D}|}K^2\tau^2,
$$

if there exists malicious clients bypassing the detection of the proposed rule. Thus, Theorem 4.2 is proved.    □

## D    PROOF OF THEOREM 4.3

*Proof.* At epoch $t$, we have the following equation:

Referring to Li et al. (2019), one has

$$
\mathbb{E}\|\boldsymbol{w}_t - \boldsymbol{w}^*\|^2 \leq (1-\eta\mu)\mathbb{E}\|\boldsymbol{w}_{t-1} - \boldsymbol{w}^*\|^2 + \eta(B+C). \tag{15}
$$

By recursively applying the inequality over successive global iterations, we have:

$$
\mathbb{E}\|\boldsymbol{w}_t - \boldsymbol{w}^*\|^2 \leq (1-\eta\mu)^{t-1}\mathbb{E}\|\boldsymbol{w}_1 - \boldsymbol{w}^*\|^2 + \eta(B+C)/(\eta\mu). \tag{16}
$$

Thus, Theorem 4.3 is proved.    □

