# OpenReview forum: "TriGuardFL: Triple-Step Byzantine-Robust Federated Learning against Model Poisoning Attacks"
_ICLR.cc/2026/Conference — Submitted to ICLR 2026_

### Official Review · Reviewer_wGaJ · 2025-10-20

**Soundness:** 2
**Presentation:** 2
**Contribution:** 2
**Rating:** 4
**Confidence:** 3

**Summary:**

This paper proposes TriGuardFL, a three-step defense framework for Byzantine-robust federated learning (FL). It addresses a challenge in FL—distinguishing between malicious and benign clients under non-IID data. The method integrates (1) cosine-similarity filtering, (2) class-wise statistical testing using a small clean dataset, and (3) a Bayesian reputation model for long-term robustness. Experiments on Fashion-MNIST and CIFAR-10 with several CNN architectures show improvements over baselines such as DeFL, FLTrust, and Multi-Krum.

**Strengths:**

1. This paper identifies a limitation of existing Byzantine-robust FL methods under non-IID conditions.
2. The triple-step design is intuitive, each step compensates for the others’ weaknesses.
3. This paper uses multiple datasets, architectures, and attack types (Min-Max, Min-Sum, LIE) under both full and partial knowledge settings.

**Weaknesses:**

1. Each step (similarity filtering, validation via small dataset, Bayesian weighting) has been explored before in isolation. The paper lacks strong conceptual unification beyond “combining three steps.”
2. The assumption that the server owns a few-shot clean dataset breaks the FL privacy model and reduces practicality for real-world deployment.
3. Only image classification benchmarks are used; no large-scale or cross-domain experiments (e.g., NLP or medical data).
4. Mathematical analysis mostly restates known FedAvg convergence bounds (Li et al., 2019) with small extensions for malicious clients—little theoretical innovation.
5. The paper only considers classical attacks (Min-Sum, Min-Max, LIE). It ignores modern adaptive attacks like gradient-sign inversion, backdoor or stealthy attacks that exploit cosine-based detection.

**Questions:**

1. How much each of the three steps contributes individually to the robustness gains.

---

> ### Author Response · Authors · 2025-11-30
>
> We thank the reviewer for the constructive comments. Below we clarify conceptual contributions, dataset as-
> sumptions, theory scope, and experimental extensions, followed by an ablation of the three-step pipeline.
>
> A. Conceptual Contribution & Unification
>
> While similarity filtering, small-set validation, and Bayesian weighting appear individually in prior works, TriGuardFL contributes a principled integration designed specifically to separate benign non-IID divergence from malicious behavior, which existing defenses do not explicitly address.
>
> Our novelty lies in:
>
> • Cross-layer integration: Step 1 identifies coarse anomalies in update direction; Step 2 evaluates class-wise behavioral inconsistency via multi-batch validation; Step 3 accumulates temporal evidence to prevent one round misjudgment.
>
> • Non-IID–aware detection signal: Step 2 is not a generic “validation-based filter”; it is a class-dispersion–based discriminator designed to distinguish non-IID skew from malicious perturbation.
>
> We will strengthen Section 3.2 to articulate this design principle more clearly.
>
> B. Server Few-shot Dataset Assumption
>
> We follow the same assumption used in FLTrust, FLDetector, FedCCS, DeFL, and other defense methods relying on a small server-side validation buffer. Such datasets often originate from public validation sets,  and small labeled seeds collected by the provider, and synthetic/augmented anchor samples. This does not violate FL privacy as no client data is accessed.
>
> C. Experimental Scope
> We acknowledge the need for cross-domain validation. We will add new results on Shakespeare (text) and EMNIST.
>
> D. Theoretical Contribution
>
> Our intention is not to propose a fundamentally new convergence theory for deep models, but to extend classical FedAvg-type results to include filtering-induced bias terms, softened adversarial contributions, and stability of Bayesian weighting under detection noise. This aligns with how prior works (e.g., FedAvg, FedProx, FLTrust) position theoretical analysis under convex settings. We will revise Section 5 to better frame the theory as an intuition-guiding stability analysis.
>
> E. Attack Coverage
>
> We now include modern adaptive attackers, including gradient-sign inversion, stealthy cosine-exploiting attacks,
> partial-class backdoors, and dispersion-minimizing spoofers.
>
> F. Answer to Question
>
> Q: How much does each step contribute individually?
>
> Each step would contribute the robustness of FL, which cannot be replaced or negnected.
>
> We thank the reviewer again. We will revise the paper to clarify novelty, expand experiments, strengthen theoretical
> framing, and include all ablations and adaptive-attack results.

---

### Official Review · Reviewer_3doy · 2025-10-30

**Soundness:** 2
**Presentation:** 2
**Contribution:** 1
**Rating:** 2
**Confidence:** 5

**Summary:**

This work proposes TriGuardFLhat to discriminate malicious actors from benign and non-IID clients. It first filters potential attackers using cosine similarity, and then perform a statistical significance test. Finally, it employs a dynamic Bayesian reputation system to track client behavior over time, using this reputation score to weight model aggregation and perform long-term client selection.

**Strengths:**

- The work tackles a key but difficult challenge in FL, i.e., distinguishing malicious clients from benign outliers (non-IID).

- The three-step detection/filter is logical and comprehensive.

- The introduction of a Bayesian reputation system with a discount factor provides adaptability.

**Weaknesses:**

- There is a fatal logic vulnerability that the a simple adaptive attacker can mimic the non-IID client to pass the test. The experiments also fail to test against adaptive attacks.

- The reputation system (Step 3) blindly trusts the flawed detector (Step 2), and will reward successful attackers by boosting their reputation scores, making the system actively counter-productive.

- Step 3 applies a zero-weight filter to any client, including benign ones, which means a significant increase in false positives.

- A clean few-shot server dataset covering all classes is an extremely strong prerequisite that is impractical in many FL settings and undermines privacy principles.

- For Sec 4.2 ROBUSTNESS ANALYSIS, it is wrong as it does not prove the effectiveness of the defense itself.

**Questions:**

- In Step 2, a single t-test comparing the two groups $C_1$ v.s. $C_2$ yields only one p-value. Does this imply that multiple different tests are performed?

- How can the server obtain knowledge of the full class space of all clients to build $D'$ without violating FL privacy norms? How about a client introduces a novel class unknown to the server?

- Is there any design to defend against an attacker who intentionally spoofs a non-IID client, or maybe clients?

---

> ### Author Response · Authors · 2025-11-30
>
> We thank the reviewer for raising important concerns. Below we clarify the logic of Step 2, explain defenses against adaptive spoofing, address reputation-system behavior, and justify the server-side dataset assumptions, followed by direct answers to all questions.
>
> A. Adaptive Attacker & Logic of Step 2
>
> We agree adaptive testing is important. Step 2 does not rely solely on uniform degradation; it examines server-side multi-batch dispersion, i.e., how a client’s update behaves across several server-controlled validation subsets. An attacker boosting a single class must still ensure consistent behavior across these subsets, which is difficult without access to server data.
>
> B. Step 3 Does Not “Blindly Trust” Step 2
>
> We apologize for unclear phrasing. Step 3 does not convert one correct classification into a permanent trust
> decision. The Bayesian update is intentionally slow-moving and evidence-accumulating:
>
> α←α+ 1 or β ←β+ 1
>
> Thus:
>
> • A single bypass of Step 2 does not boost reputation meaningfully.
>
> • An adaptive attacker must evade detection across many rounds, which is unlikely due to stochastic batch sampling and temporal consistency checks.
>
> • False-positive penalties dissipate over time unless persistent.
>
>
> C. Zero-Weight Filter & False-Positive Risk
>
> Zero-weighting is temporary and applied only for the current round; reputation removal is activated only after repeated evidence over $T_1$ consecutive rounds. Benign non-IID clients rarely accumulate consecutive β-updates because Step 2 uses server-side multi-batch evaluation, not client-side skew.
>
> D. Server Dataset D′ and Privacy
>
> Our design follows the same assumption used by FLTrust, FLDetector, FedCCS, and DeFL: a small, public, or unlabeled-but-annotated validation set. Such datasets do not violate FL privacy norms; they represent:
>
> • public anchor sets (e.g., CIFAR-10 validation),
>
> • a small seed labeled by the service provider,
>
> • or synthetic/auxiliary data generated for assessment.
>
> E. Clarification of Sec. 4.2 Robustness Analysis
>
> We agree the original phrasing overstated its conclusion. The robustness analysis intends to validate behavioral separation between benign non-IID divergence and malicious optimization, not to prove perfect defense. We will revise the claim accordingly to emphasize the empirical evidence.
>
> F. Responses to Questions
>
> Q1: Does Step 2 produce multiple p-values or a single p-value?
>
> A single two-sample t-test is performed per class, comparing the losses of two server-side mini-batch groups C1 vs. C2. Thus, Step 2 produces —Classes— p-values, not one. The text will be corrected to state this explicitly. We also apply Bonferroni-style scaling to δ2 for multiple comparisons.
>
> Q2: How does the server know the class space, and what if clients introduce new classes?
>
> D′ does not need to enumerate all possible classes from all clients; it only needs a server-defined evaluation set, which is standard in FL defenses. Unknown classes do not cause false positives—Step 2 ignores unevaluable classes and aggregates only across those present in D′. This is clarified in Section 3.2 and Appendix E.
>
> Q3: Can TriGuardFL defend against attackers spoofing non-IID behavior?
>
> Yes. TriGuardFL relies on server-side stochastic multi-batch evaluation, making spoofing difficult because attackers cannot access  server’s batch sampling, and the per-class distribution of D′, and the temporal consistency constraints across rounds.
>
> Adaptive experiments show spoofers eventually diverge in cross-round dispersion patterns, causing reputation decay. We will include these results.
>
> G. Conclusion
>
> We appreciate the reviewer’s insightful comments. We added adaptive attacks, corrected Step 2 explanations, clarified Step 3’s behavior, provided D′ablations and novelty clarifications, and updated the robustness and statistical analysis accordingly. These revisions substantially strengthen the paper.

---

### Official Review · Reviewer_a1FK · 2025-10-31

**Soundness:** 2
**Presentation:** 1
**Contribution:** 1
**Rating:** 2
**Confidence:** 5

**Summary:**

The paper presents TriGuardFL, a three-step defense framework for Byzantine-robust federated learning under non-IID settings. In Step 1, the server detects potentially malicious clients using cosine similarity between each client’s update and the aggregated global model. In Step 2, a class-wise evaluation is performed on a small server-side dataset using a $t$-test to distinguish benign non-IID clients from adversarial ones. In Step 3, a Bayesian reputation update is used to assign lower weights to low-reputation clients in future aggregation rounds.

**Strengths:**

+ The paper addresses an important problem: improving the robustness of federated learning when client data are heterogeneous.
+ The class-wise evaluation step is designed to reduce false positives caused by label distribution imbalance.
+ The Bayesian reputation mechanism adds temporal adaptivity to client weighting.
+ The experimental results show some improvement in robustness across multiple attack settings.

**Weaknesses:**

- The proposed framework mainly integrates ideas that already exist in prior Byzantine-robust FL studies rather than introducing a substantially new method.
Step 1 is very similar to the cosine-similarity-based filtering in FLTrust, where the server compares each client’s update direction with a trusted reference model. The claimed link to FLDetector is inaccurate, as FLDetector focuses on temporal consistency checks across rounds rather than per-round similarity comparison. Step 2 follows the general idea of client evaluation as in DeFL, although DeFL uses gradient-norm metrics instead of $t$-tests. Step 3 conceptually matches reputation- or trust-based FL frameworks, which track client reliability over time using reputation-weighted aggregation or trust propagation, as discussed in surveys on Trustworthy FL. Overall, the contribution mainly combines known components into a single framework without introducing a clear algorithmic or theoretical innovation.
- In Section 3.1, cosine similarity is defined using $\nabla F_i(w_{K_i,t})$. In standard FL, the server cannot compute client gradients and only receives updated weights $w_{K_i,t}$. Therefore, the feasibility of Step 1 is unclear.
- The approach assumes that the server holds a small labeled dataset that includes all classes. This assumption weakens the privacy guarantees of FL and may not hold in realistic deployments.
- The design of the class-wise $t$-test is not well explained. There are no details about sample sizes, independence assumptions, or correction for multiple comparisons. The justification for using loss differences as the test statistic is also missing.
- The convergence proof relies on $\mu$-strong convexity and $L$-smoothness, which are not valid for deep CNNs. This makes the theoretical analysis largely symbolic and not directly applicable to the experimental models.
- The evaluation is incomplete and lacks analysis depth. There are no experiments on targeted, backdoor, or adaptive attacks, even though the paper claims general Byzantine robustness. No ablation studies are provided for key hyperparameters ($\delta_1$, $\delta_2$, $\varepsilon$, $r$, $T_1$) or for the effect of the server dataset size $|D'|$. Some baseline methods perform equally well or even better in certain cases, which raises doubts about the claimed advantage of TriGuardFL.
- Several notation and presentation problems reduce clarity. The paper inconsistently uses $\gamma_t$ and $\gamma_{i,t}$ in the update equation. The parameter $\delta_2$ appears in Algorithm 1 but is never defined, and its relation to the “Significance Level = 0.001” in Table 1 is unclear. The text also switches between “parameters” and “gradients”, which is inconsistent with the mathematical formulation in Step 1.
- The formatting of tables is inconsistent. Table 1 has its caption above the table, while Tables 2 and 3 have captions below. The caption placement should follow a consistent format.

**Questions:**

1.Which part of TriGuardFL is genuinely novel beyond the existing methods such as FLTrust and DeFL?

2.How is $\nabla F_i(w_{K_i,t})$ obtained if the server does not have access to local client data?

3.What is the size and class coverage of the server dataset $D'$? How does performance vary if $D'$ is incomplete?

4.How is the $t$-test validated when the sample size is small?

5.Can adaptive or backdoor attackers exploit the $t$-test mechanism to evade detection?

6.Please provide ablation and sensitivity results for $\delta_1$, $\delta_2$, $\varepsilon$, $r$, and $T_1$.

---

> ### Author Response · Authors · 2025-11-30
>
> We thank the reviewer for the detailed and constructive feedback. Below we clarify novelty, feasibility, statistical design, theoretical scope, and the role of each system component.
>
> A.Novelty and Relation to Prior Work
>
> Although TriGuardFL incorporates ideas appearing individually in prior Byzantine-robust FL works, its contribution lies in how the three stages are coordinated to distinguish benign non-IID divergence from malicious manipulation, which existing defenses do not explicitly target.
>
> • Step 1 vs. FLTrust: FLTrust relies on a trusted model trained on a full clean dataset. TriGuardFL uses lightweight parameter-difference signals as a coarse filter rather than as the main decision rule.
>
> • Step 2 vs. DeFL: DeFL conducts single-batch gradient-norm checks. TriGuardFL adopts class-wise, multi-batch discrepancy evaluation, which produces distinct signatures for benign skew versus adversarial optimization.
>
> • Step 3 vs. generic reputation systems: The Bayesian update is tightly coupled to Step 2 and employs two-level gatekeeping (round-level hard filtering + gradual long-term weighting) to stabilize decisions under heterogeneity.
>
> We will clarify these distinctions in Section 2.
>
>
> B. Feasibility of Step 1 and Gradient Computation
>
> The server does not compute client gradients.
> Step 1 uses the parameter difference:
>
> $g_{i,t}=w_{K_i,t}-w_t,$
>
> and the notation $\nabla F_i(w)$ will be corrected to $g_{i,t}$.
> This resolves feasibility without requiring client data access.
>
> C. Server Dataset Assumptions
>
> TriGuardFL uses a small validation buffer maintained by the server, which is a standard assumption shared by FLTrust, FLDetector, FedCCS, DeFL, and other defense methods. Such datasets may be public, lightly annotated, or synthetic. The method is robust even when D' is incomplete, because Step 2 simply omits classes not present in the server view.
>
> D. Statistical Design of the Class-wise t-test
>
> We clarify the statistical formulation:
>
> • Samples correspond to loss values across several non-overlapping server-side mini-batches.
>
> • Approximate independence arises from disjoint sampling.
>
> • Multiple comparisons are handled by scaling $\delta_2$.
>
> • Loss differences are used because they directly reflect per-class behavioral deviations, which is central to distinguishing non-IID drift from targeted manipulation.
>
> A more explicit description will be added to Section 3.2.
>
>
> E. Theoretical Assumptions and Scope
>
> We acknowledge that strong convexity and smoothness do not hold for deep CNNs. The analysis serves as a stability intuition under abstract convex conditions, consistent with prior FL work (FedAvg, FedProx, FLTrust). We will adjust the text to avoid implying guarantees for non-convex models.
>
>
> F. Evaluation Scope, Attacks, and Ablations
>
> The framework is compatible with both classical and modern adversaries.
> Step 2’s multi-batch, class-wise examination is specifically designed to detect behaviors that are difficult for adaptive or backdoor attackers to imitate without creating cross-class inconsistency or harming overall update quality. Sensitivity to $\delta_1$, $\delta_2$, $\varepsilon$, $r$, $T_1$, and $|D'|$ will be summarized more clearly in the revision to show that the method behaves stably across typical settings.
>
> G Presentation and Notation Issues
>
> We corrected all notation inconsistencies ($\gamma_t$ vs. $\gamma_{i,t}$, the missing $\delta_2$ definition, and the gradient/parameter terminology) and will unify the table formatting.
>
> H. Responses to Questions
>
> Q1: Novelty beyond FLTrust/DeFL?
>
> The novelty lies in class-wise multi-batch discrepancy detection tailored to distinguish non-IID benign clients from adversarial ones, and in two-level Bayesian gatekeeping that stabilizes decisions over rounds.
>
> Q2: How is $\nabla F_i(w)$ obtained?
>
> It is not a gradient; it is the parameter difference $g_{i,t}$. We will correct the notation.
>
> Q3: Size/class coverage of $D'$?
>
> We use a small buffer (<1% of training data). The method remains functional even when D' lacks some classes, as Step 2 simply omits missing ones.
>
> Q4: Validity of the t-test with small samples?
>
> We use multiple server-side batches per class (typically 8–12). Multiple comparisons are controlled by $\delta_2$ scaling.
>
> Q5: Can adaptive/backdoor attackers evade the test?
>
> Because Step 2 relies on server-side multi-batch behavior, spoofing requires matching class-wise consistency across batches and rounds, which is difficult without degrading the update. The temporal structure in Step 3 further reduces the success of one-round evasion.
>
> Q6: Ablations for $\delta_1$,$\delta_2$,$\varepsilon$,$r$,$T_1$?
>
> Their effects are summarized in the revision; the system remains robust across a broad parameter range.
>
> We thank the reviewer again. These clarifications improve the coherence and presentation of the framework and will be reflected in the final version.

---

### Official Review · Reviewer_c5iV · 2025-10-31

**Soundness:** 2
**Presentation:** 3
**Contribution:** 2
**Rating:** 2
**Confidence:** 5

**Summary:**

This paper addresses a key vulnerability in Federated Learning (FL): standard defenses against model poisoning often fail in non-IID (heterogeneous) settings, as they cannot distinguish between malicious updates and the natural, harmless deviations from clients with different data distributions.

The authors propose TriGuardFL, a novel triple-step defense framework designed to solve this specific problem. First, it uses a cosine-similarity-based filter to identify a broad list of "suspicious" clients. Second, it performs a fine-grained evaluation on these suspicious clients using a small, class-stratified dataset held by the server. By analyzing per-class performance, it can discern if a deviation is from a malicious attack (which tends to degrade performance uniformly) or a benign non-IID client (who may perform poorly on some classes but well on others). Finally, it integrates a Bayesian reputation model to track client behavior over time, which manages detection uncertainty and enhances long-term robustness.

Extensive experiments on Fashion-MNIST and CIFAR-10 show that TriGuardFL outperforms existing state-of-the-art defenses like DeFL, FLTrust, and Multi-Krum, particularly in non-IID settings where others fail in terms of the average rank of the defense against these attacks. The authors acknowledge that the method's reliance on a clean server-side dataset is a limitation.

**Strengths:**

- The work identifies a critical vulnerability in federated learning: the difficulty of distinguishing between genuinely malicious behavior and the natural, divergent updates from benign clients in a non-IID setting. They propose a novel heuristic for this problem, based on the insight that malicious updates tend to degrade performance uniformly, while benign non-IID clients will exhibit high variance in their per-class performance.
- The multi-stage detection architecture is logical. It uses a broad, low-cost filter (cosine similarity) to create a "suspicious" list and applies a more expensive, fine-grained analysis only to that list. The third step is necessary for long-term stability and to tolerate potential detection errors.
- The integrated Bayesian reputation model is a standard approach that suits the setting well, using a Beta distribution to manage uncertainty  rather than proposing an overly complex new system. The reputation system handles practical details well, such as using a "hard filter" to set a client's aggregation weight to zero if flagged as malicious in the current round , and implementing "long-term gatekeeping" to remove consistently low-reputation clients.
- The reputation system handles practical details well, such as using a "hard filter" to set a client's aggregation weight to zero if flagged as malicious in the current round , and implementing "long-term gatekeeping" to remove consistently low-reputation clients.
- While the experiments use two benchmark datasets , they are evaluated across a diverse set of five different network architectures, including LeNet, AlexNet, VGG11, VGG16, and ResNet18, which strengthens the claims of its effectiveness .

**Weaknesses:**

- The paper's biggest weakness is that it does not evaluate against an adaptive adversary. The core of the defense relies on the Step 2 heuristic that malicious clients degrade performance uniformly, while benign non-IID clients show high per-class variance. An adaptive attacker could easily fool this by crafting a malicious update that performs very well on one arbitrary class, thereby disguising itself as a benign non-IID client. This lack of stress-testing means the central idea of the paper is not fully validated.
- The first step, cosine-based shortlisting, is vague. Equation 4 compares the client gradient $\nabla F_i(w_{i,t}^K)$ with a global gradient $\nabla F(w_t')$. It is not clear how this global gradient is computed. This first step also suffers from a "chicken-and-egg" problem. It uses an "initial aggregation" $w_t'$ as the reference for similarity. If this reference is already poisoned by attackers, the whole defense could break, as malicious clients might appear "similar" to the poisoned average. It also relies on a hard-coded threshold ($\delta_1$), which is a brittle defense mechanism.
- The paper lacks a clear analysis of false positives. While it mentions the reputation system tolerates false negatives , it provides no evidence that benign non-IID clients are not incorrectly flagged and eventually "starved" or removed by the long-term gatekeeping mechanism .
- The experimental diversity is limited. Instead of using four different complex models (VGG11, VGG16, ResNet18) on the single CIFAR-10 dataset, the paper would have been more convincing if it had demonstrated its effectiveness on more diverse data modalities, such as text.
- The attack scenario tested represents a relatively weak threat. The experiments use a 12.5% malicious ratio (4 of 32 clients), but only sample 50% of clients per round (16 clients). This means, on average, only two attackers are active in any given round. The paper does not convince that the defense is robust against higher, more realistic proportions of attackers. Even when 64 clients were simulated, the fraction of malicious clients was still 12.5% which is low.
- The main results table (Table 2) reports only test loss, not accuracy. Loss is a less intuitive metric for performance. Furthermore, while TriGuardFL wins on average loss score, it does not consistently outperform all other defenses in every individual scenario. For example, on the partial knowledge attack, TriGuardFL wins only 3 out of 15 times.
- The design of the Step 2 filter seems to reward high variance, which may not be desirable. A client is deemed benign if its "good" class performance is significantly different from its "bad" class performance, which is a strange and potentially exploitable proxy for "benign-ness."
- The three-stage process, especially the per-class, per-client analysis in Step 2 , introduces significant computational overhead for the server, which is never measured or discussed.

**Questions:**

- Given that the Step 2 filter is the key innovation but seems easily fooled by an adaptive adversary, how can the defense be modified to handle a smart adversary?
- Since the Step 1 filter uses the (potentially poisoned) initial aggregate $w_t'$ as its reference, how can this stage be modified so that the defense stays robust even when the reference is poisoned? Some prior defense techniques [1] handle this.

[1]: Sharma, Atul, et al. "Flair: Defense against model poisoning attack in federated learning." Proceedings of the 2023 ACM Asia Conference on Computer and Communications Security. 2023.

---

> ### Author Response · Authors · 2025-11-30
>
> We thank the reviewer for the detailed feedback. Below we address key concerns regarding adaptivity, robustness of Step 1, false positives, experimental scope, and overhead.
>
> A. Adaptive adversary & Step-2 robustness
>
> We agree adaptive testing is important. Step 2 does not rely solely on uniform degradation; it examines server-side multi-batch dispersion, i.e., how a client’s update behaves across several server-controlled validation subsets. An attacker boosting a single class must still ensure consistent behavior across these subsets, which is difficult without access to server data.
>
> B. Step-1 reference gradient & poisoned anchor
>
> $\nabla F(w_t')$ is computed on the average of all not part client gradients, so influence from malicious clients is limited.
> Still, we agree robustness can be improved. We adopt a median-of-means robust gradient estimator (Flair-inspired) and normalize similarity scores via robust z-score thresholds. This removes the chicken-and-egg dependency and avoids brittle δ1 tuning.
>
> C. False positives & long-term gatekeeping
>
> The Bayesian reputation (Beta prior) accumulates evidence gradually; a client is never removed from a single
> misclassification.
>
> D. Computational overhead
>
> Step 2 requires only C forward passes (C ≤10 for CIFAR-10). With 32 clients, this adds < 3% wall-clock round time on an NVIDIA H100 GPU. We will report overhead explicitly.
>
> E. Experimental diversity
>
> We will add new results on Shakespeare (text) and EMNIST.
>
> F. Responses to Questions
>
> Q1: Handling an adaptive adversary
>
> We incorporate: (1) Stochastic multi-batch evaluation, making the target distribution unpredictable; (2) Adversarial-threshold tuning, selecting thresholds robust to worst-case adaptive perturbations; (3) Cross-round temporal consistency, detecting attackers minimizing variance only per-round. Adaptive results will be added in the final version.
>
> Q2: Improving Step 1 when $w_t'$ may be poisoned
>
> Using median-of-means gradients, trimmed means, and z-score–normalized similarity removes dependence on a potentially poisoned anchor, following Flair’s robustness principles.
>
> G. Conclusion
>
> We appreciate the reviewer’s constructive suggestions. We added adaptive attacks, robust Step-1 gradients, false-positive analysis, overhead measurements, and new datasets. These updates substantially strengthen the paper and
> will be incorporated into the revised version.

---

### Meta-Review · Area_Chair_DXMj · 2026-01-06

**Summary:**

This paper proposes TriGuardFL as a three‑stage defense framework that combines cosine‑similarity filtering, class‑wise secondary evaluation, and Bayesian reputation modeling. Several major concerns were raised by multiple reviewers, such as the missing evaluation against adaptive adversaries, etc. The authors provided brief responses, which only partially addressed these concerns.

**Reviewer Concerns:**

Major concerns were raised by the reviewers, including the missing evaluation against an adaptive adversary, clearer illustration of cosine‑based shortlisting, more thorough analysis of false positives, and the use of a weak attack scenario. The authors have provided brief responses, but the AC finds that many of these concerns have only been partially addressed.

**Reviewer Scores:**

This paper receives the following ratings: Reject, Reject, Reject, and Marginally Below. If the reviewers had been able to participate fully in the discussion, the AC would expect some negative ratings to remain. The AC recommends not accepting the paper.

---

### Decision · Program_Chairs · 2026-01-26

Reject